# Characterization and Mapping of a Novel Premature Leaf Senescence Mutant in Common Tobacco (*Nicotiana tabacum* L.)

**DOI:** 10.3390/plants8100415

**Published:** 2019-10-15

**Authors:** Xiaoming Gao, Xinru Wu, Guanshan Liu, Zenglin Zhang, Jiangtao Chao, Zhiyuan Li, Yongfeng Guo, Yuhe Sun

**Affiliations:** 1Tobacco Research Institute, Chinese Academy of Agricultural Sciences, Qingdao 266101, China; gaoxiaoming@caas.cn (X.G.);; 2Plant Genetic, Gembloux Agro-Bio Tech, University of Liege, Gembloux B-5030, Belgium

**Keywords:** tobacco, EMS mutagenesis, premature leaf senescence, SSR markers, plant hormone

## Abstract

As the last stage of plant development, leaf senescence has a great impact on plant’s life cycle. Genetic manipulation of leaf senescence has been used as an efficient approach in improving the yield and quality of crop plants. Here we describe an ethyl methane sulfonate (EMS) mutagenesis induced premature leaf senescence mutant *yellow leaf 1* (*yl1*) in common tobacco (*Nicotiana tabacum* L.). The *yl1* plants displayed early leaf yellowing. Physiological parameters and marker genes expression indicated that the *yl1* phenotype was caused by premature leaf senescence. Genetic analyses indicated that the *yl1* phenotype was controlled by a single recessive gene that was subsequently mapped to a specific interval of tobacco linkage group 11 using simple sequence repeat (SSR) markers. Exogenous plant hormone treatments of leaves showed that the *yl1* mutant was more sensitive to ethylene and jasmonic acid than the wild type. No similar tobacco premature leaf senescence mutants have been reported. This study laid a foundation for finding the gene controlling the mutation phenotype and revealing the molecular regulation mechanism of tobacco leaf senescence in the next stage.

## 1. Introduction

Tobacco (*Nicotiana tabacum* L.) is an important agricultural crop widely grown in the world. It plays an important role in plant genetics and transgenic research as a model plant [1,2]. Also, it can be used as a bioreactor for vaccine production [3,4,5]. In addition, due to the huge biomass of tobacco, it has become a potential bioenergy plant [6,7,8]. In tobacco production, mature/ripe leaves are harvested, so leaf senescence is critical for both the yield and quality of tobacco products. Therefore, studying tobacco leaf senescence has important theoretical significance and a practical application value. Although plant leaf senescence regulation has been extensively studied in model systems such as Arabidopsis and rice, limited information is available about tobacco leaf senescence regulation at present. Several studies focused on transcriptome and metabolome of tobacco leaf senescence [9,10,11,12]. These studies have provided global and potential gene dynamic expression changes and signaling pathways for leaf senescence regulation in tobacco, but few genes involved in regulation of tobacco leaf senescence have been identified.

Senescence is the terminal phase of plant development that typically displays as a decline at the cell, tissue, organ, and whole plant levels [13]. Leaf senescence is the main part of plant senescence, and is strictly controlled at the molecular level during plant development [14]. In a controlled environment with sufficient nutrition and favorable growth conditions, leaf senescence depends on the leaf age and development stage of the plant [15,16]. Leaf senescence is affected by both endogenous and exogenous factors, which regulate the expression of genes involved in the execution process of senescence, and eventually lead to the death of the whole plant [17]. Furthermore, leaf senescence is regulated by a complex regulatory network composed of transcription factors, hormones, environmental factors, sugar metabolism, epigenetic regulation, and miRNA mediated regulation [17,18,19,20,21]. Leaf senescence is not a completely passive and adverse biological process. During leaf senescence, macromolecules such as proteins, lipids, and nucleic acids are broken down. The nutrients released from the catabolism of these macromolecules, as well as other nutrients, are recycled to new buds, young leaves, developing fruits, and seeds, which influence the yield of crops [15]. Premature leaf senescence generally causes significant yield loss while delayed leaf senescence leads to increased grain production in many crop species [22,23].

Chlorophyll breakdown is one of the most important catabolic processes during leaf senescence in higher plants [24]. The yellowing of leaves caused by chlorophyll loss is the major phenotype of plant senescence. Total chlorophyll content is the most commonly used parameter for characterizing senescence progression. Drastic changes in gene expression are associated with leaf senescence. Arabidopsis leaf senescence transcriptome data revealed that many transcription factor families are overrepresented in the transcriptome of leaf senescence [18,25]. In addition to transcription factors, many genes associated with metabolism and signal transduction were identified to be associated with leaf senescence. Based on their different expression patterns, genes with differential expression were classified as senescence down-regulated genes (SDGs), such as *ribulose bisphosphate carboxylase small subunit* (*RBCS*) encoding a member of the rubisco small subunit, a key enzyme in photosynthesis, and senescence up-regulated genes, which are also called senescence-associated genes (SAGs) [26]. While many of the *SAGs* could be induced by senescence as well as other senescence-inducing factors, such as environmental stresses, *SAG12* was found to be strictly induced by natural senescence [27], and thus, often used as a leaf senescence marker gene. Plant hormones are important endogenous factors involved in the regulation of leaf senescence. Some of the senescence regulating plant hormones are also involved in the plant’s response to environmental stresses and stress-induced senescence [19,28,29]. Thus, leaf senescence often involves complex cross talks between signaling pathways associated with multiple hormones and stress signals.

Molecular markers are powerful tools for genetic research and plant breeding that have been widely used in many types of biological research. Genetic linkage mapping based on molecular markers can clarify the structure and organization of the genome. In tobacco, a large number of simple sequence repeat (SSR) markers have been developed based on the released data of tobacco genome sequences and expressed sequence tags. Bindler et al. [30,31] used two different tobacco varieties, Hicks Broadleaf and Red Russian, in constructing a mapping population, and they published more than 5000 SSR markers and obtained a high-density tobacco genetic map containing 24 linkage groups (LGs) with 2363 SSR markers loci. Tong et al. [32] developed a total of 4886 SSR markers (including 1365 genomic SSRs and 3521 EST-SSRs), which were functional in a set of eight tobacco varieties of four different types and were essentially non-overlapping with the set published by Bindler et al. [30,31]. These SSR markers have been wildly used for the identification of quantitative trait loci (QTLs) associated with tobacco disease resistance [33,34,35,36] and tobacco traits [37], and they have also been used for genetic diversity analysis of tobacco varieties [38,39].

In this study, we identified a tobacco mutant *yl1*, which exhibited a premature leaf senescence phenotype. We characterized the premature leaf senescence phenotype of *yl1*, determined the physiological parameters related to leaf senescence and the expression of senescence marker genes, analyzed its phenotypic inheritance, and mapped the causal gene. 

## 2. Results

### 2.1. The yl1 Plants Exhibit Premature Leaf Senescence

Compared with HonghuaDajinyuan (HD), *yl1* plants showed a premature leaf senescence phenotype from about 50 days after transplanting (DAT), and throughout the subsequent developmental stage (Figure 1). No significant difference of leaf color was observed between HD and *yl1* 35 DAT or at earlier development stages (Figure 1A). The yellowing of the bottom leaves appeared in the *yl1* plants around 50 DAT, and the leaf color was lighter than HD (Figure 1B). At the flowering stage, about 75 DAT, significant difference in leaf color of the bottom and middle leaves were observed between the *yl1* and HD plants (Figure 1C,E) and this difference in leaf yellowing was more profound at 95 DAT when the upper leaves of the *yl1* plants became completely yellow, whereas the majority of leaves in the HD plants still retained their green color (Figure 1). These results suggest that the *yl1* plants seemed to display a phenotype of premature leaf senescence.

### 2.2. Alterations of Leaf Senescence Related Parameters

In order to further characterize the phenotype of *yl1*, we determined the chlorophyll content, *F*v/*F*m ratio, soluble protein content, and the expression of senescence marker genes in *yl1* in comparison with HD (Figure 2). Consistent with the premature leaf yellowing phenotype, Chlorophyll *a* (Chl *a*), Chlorophyll *b* (Chl *b*), and the total chlorophyll (Chls) of the middle leaves in *yl1* were lower than in counterpart leaves of the HD plants at the same developmental stages. The contents of Chl *a*, Chl *b*, and Chls decreased rapidly in *yl1* 50 DAT, whereas the HD plants did not start losing chlorophyll until 75 DAT (Figure 2A–C). In tobacco, senescence of leaves progresses from the bottom to the top of a plant. At 75 DAT, although no significant difference in chlorophyll contents was observed in upper leaves, chlorophyll contents in the middle and lower leaves were significantly lower in *yl1* than in the HD plants (Figure 2D–F). At 75 DAT, the *F*v/*F*m ration and soluble protein in upper, middle, and lower leaves of *yl1* were significantly lower than in the HD plants (Figure 2G,H). Furthermore, the expression of the marker genes *SAG12* and *RBCS* of the individual samples was determined. As shown in Figure 2I and J, in *yl1*, expression of the senescence marker gene *SAG12* was first detected at 50 DAT and the photosynthetic gene *RBCS* became undetectable at 75 DAT. For HD plants, *SAG12* expression was first detected at 75 DAT and *RBCS* expression became undetectable at 95 DAT. Significantly lower *SAG12* and higher *RBCS* expression were detected at different leaf positions of 75 DAT plants (Figure 2K,L). The premature leaf senescence phenotype of *yl1* was well reflected in the expression patterns of *SAG12* and *RBCS*.

### 2.3. The Premature Leaf Senescence of yl1 is Controlled by a Single Recessive Gene

To analyze the inheritance of the premature leaf senescence phenotype of *yl1*, two wild type tobacco varieties, HD and Gexin 3 (G3), were crossed with *yl1*. All the F1 plants showed a normal green phenotype similar to the wild type. A total of 43 out of 154 (HD × *yl1*) and 40 out of 155 (G3 × *yl1*) F2 plants, respectively, displayed the premature leaf senescence phenotype, showing a segregation ratio of 3:1 (χ^2^ < χ^2^_0.05_ = 3.841; Table 1). These results suggest that the premature leaf senescence phenotype of *yl1* was controlled by a single recessive gene. To further verify this observation, BC1F1 populations were generated by crossing the F1 plants of HD × *yl1* and G3 × *yl1* with *yl1* separately. The phenotypic segregations of the BC1F1 populations correlated to the expected ratio of 1:1 with similar numbers of wild type and mutant type plants (χ^2^ < χ^2^_0.05_ = 3.841; Table 1).

### 2.4. Preliminary Mapping of YL1

The BC1F1 population developed between parent G3, and *yl1* was used to map *YL1*. A total of 265 simple sequence repeat (SSR) markers showing polymorphism between G3 and *yl1* were identified from 1376 pairs of SSR markers by Bindler et al. [30,31] and Tong et al. [32], from which 96 markers were evenly distributed on 24 linkage groups (LGs) of tobacco and were selected to screen 19 recessive individuals from the BC1F1 population of G3 and *yl1*. The results showed that the marker PT53066 located on LG11 was associated with the *yl1* phenotype (Figure 3A). In order to obtain more linked markers, we used more polymorphic SSR markers on the flanks of PT53066 to screen the 19 mutant plants and identified three more linked markers: PT60305, PT60998, and PT60975 (Figure 3B). In order to obtain the exact location of the *YL1* gene on LG11, we enlarged the BC1F1 population to get 46 premature leaf senescence individuals and screened them with the above four linked SSR markers. The isolation of each SSR marker and *YL1* was converted into genetic distance by the QTL Ici Mapping software (V4.0). The results showed that *YL1* was located between the markers PT53066 and PT60305, with the genetic distance of 1.08 and 3.51 cM, respectively (Figure 3B).

### 2.5. Plant Hormone Treatments

In order to find out whether *YL1* is involved in hormone signaling pathways that regulate leaf senescence, leaf discs from the middle leaves of 50 DAT plants were used in plant hormone treatments (Figure 4). Without plant hormone treatment, the yellowing of the *yl1* leaves was faster than that of HD (Figure 4 Control). As senescence-promoting hormones, treatments of both 1 - aminocyclopropane - 1 - carboxylic acid (ACC) and methyl jasmonate (MeJA) accelerated yellowing of leaf discs with a significantly more profound effect on the *yl1* leaves, suggesting that *yl1* was more sensitive to ACC and MeJA than the HD plants (Figure 4).

## 3. Discussion

The *yl1* mutant has been continuously planted in Zhucheng City, Shandong Province, China, for four years. A reliable and stable early yellowing phenotype has been consistently observed in *yl1* plants over the years. The external performance of leaf senescence is that the leaf color turns yellow or even dead, while the internal performances include the chloroplast structure changing significantly or even completely disintegrating, and the decrease of the photosynthesis and protein [15]. One of the early physiological manifestations of leaf senescence is the collapse of the membrane structure due to membrane lipid de-esterification [40]. With the occurrence of senescence, the outer layer of the chloroplast disappears, and the chlorophyll degrades. Chlorophyll *a* (Chl *a*) and Chlorophyll *b* (Chl *b*) are the two prevailing forms of chlorophyll, which are involved in different light harvesting. Chl *a* is associated with the energy processing centers of the photosystems, while Chl *b* is considered to be an accessory pigment that transfers light energy to Chl *a* [41]. The chlorophyll breakdown process is initiated by conversion from Chl *b* to Chl *a* by a Chl *b* reductase [24]. The content of Chl *a*, Chl *b*, and total chlorophyll (Chls) in *yl1* was lower than that in HD during the whole development process, and the difference was more significant at the middle and late stages of development. Moreover, chlorophyll degradation was initiated earlier in *yl1* than in HD at an early development stage of than 50 DAT (Figure 2A–C). Therefore, we believe the low chlorophyll content at late stage is due to early chlorophyll degradation, a typical premature senescence phenotype. The phenotype of the *yl1* mutant was further confirmed by the difference in *F*v/*F*m ratio, soluble protein content, and expression levels of the senescence-associated marker genes (Figure 2G–L).

Mapping functional genes using EMS mutants combined with map-based cloning has been applied in many crops; however, in tobacco, it has lagged behind when compared with those of other plants such as rice. The common tobacco (*Nicotiana tabacum* L.) is an allotetraploid (2n = 48) species produced by the hybridization and chromosome doubling of *N. sylvestris* (2n = 24) and *N. tomentosiformis* (2n = 24) [42], with 4.5 Gb genome and more than 70% repetitive sequences [43]. The difficulty of mapping tobacco genes by map-based cloning is partially due to the large genome of tobacco with lots of repetitive sequences. Wu et al. (2014) used SSR markers by Bindler et al. [30,31] and Tong et al. [32] to perform a preliminary mapping of a light color mutant *ws1* controlled by two recessive nuclear genes. The *ws1a* locus was mapped between PT54006 and PT51778 with a genetic distance of 8.04 and 3.96 cM, respectively, and the *ws1b* locus was mapped between PT53716 and TM11187, each with a genetic distance of 8.56 cM from *ws1**b* [44]. Michel et al. [45] located *NtTPN1*, a gene required for potato virus Y-induced veinal necrosis in tobacco, between PT60530 and PT61143 with a distance of 0.5 and 2.9 cM, respectively. A tobacco black shank resistance gene *Ph* was located at the top of LG20, but there were no more PT SSR markers available for subsequent localization [46]. The gene responsible for the premature senescence phenotype of *yl1* was mapped between the markers PT53066 and PT60305, and the genetic distance was 1.08 and 3.51 cM, respectively. Similarly, we were not able to further narrow down the interval using the PT/TM SSR markers. Obviously, we need more molecular markers to enrich the linkage genetic map of tobacco.

Recently, a total of 1,224,048 non-redundant *Nicotiana* SSR markers were discovered and characterized in silico in seven *Nicotiana* genomic sequences, of which 99.98% are novel [47]. In addition to SSR markers, single nucleotide polymorphisms (SNPs) are also commonly used molecular markers [48]. With the rapid advance in next generation sequencing (NGS) technologies, it has become possible to complete genome sequencing of large crop species. Especially for tobacco, which is a crop with huge genome and low genetic diversity, it is necessary to develop SNPs from the genomic region due to their wide distribution in the genome. In the past five years, four sets of tobacco SNPs have been developed in tobacco through different varieties, sequencing populations, and sequencing methods; they have been integrated into the 24 LGs developed by Bindler et al., and new genetic linkage maps have been drawn [49,50,51,52]. These will be good tools for the fine mapping of *YL1* in future.

In view of the important role of plant hormones in leaf senescence, we attempted to explore the relationship between the causal gene and hormone pathways by the hormone treatment of detached leaves, and to provide potential methods for screening candidate genes in the future. Ethylene is one of the most important plant hormones that induce leaf senescence. Some ethylene receptors, such as EIN2, a core member of ethylene signaling pathway, play a role in other hormone signaling and act as a node in the interaction between plant hormone signals [19]. Jasmonic acid induces the expression of key enzymes involved in chlorophyll degradation, so it has long been thought to accelerate leaf senescence [53]. In addition, jasmonic acid regulates leaf senescence by interacting with other hormonal signaling pathways such as salicylic acid, auxin, and ethylene [54,55,56]. When HD and *yl1* were treated with ACC and MeJA, both of them had accelerated leaf senescence. It is clearly that *yl1* is more sensitive to ACC and MeJA. However, we did not find that *yl1* showed more interesting responses to ethylene and jasmonic acid.

In summary, we have identified a novel tobacco EMS mutagenesis induced premature leaf senescence mutant controlled by a recessive single gene and have preliminarily mapped the causal gene to an interval between PT53066 and PT60305 on the tobacco LG11, with the genetic distance of 1.08 and 3.51 cM, respectively. This provides good genetic material. Further cloning and characterization of the causal gene will lead to a better understanding of the molecular mechanisms underlining leaf senescence in tobacco and provide useful information in the genetic manipulation of tobacco leaf senescence.

## 4. Materials and Methods 

### 4.1. Plant Materials and Genetic Populations

The *yl1* mutant was identified from the chemical mutagen EMS induced library derived from the common tobacco variety HonghuaDajinyuan (HD) [57]. The *yl1* mutant was identified from one the M2 lines, and further verified in the M3 and M4 generation. HD and another wild type tobacco variety G3 (Gexin 3) were crossed with *yl1*, respectively, to produce genetic populations. The F1 (HD × *yl1*; G3 × *yl1*) plants were self-pollinated to generate F2 segregating populations. BC1F1 segregating populations ((HD × *yl1*) × *yl1*; (G3 × *yl1*) × *yl1*) were generated by crossing F1 with *yl1*. All the plants were grown in a field in Zhucheng City, Shandong Province. Leaf senescence phenotypes were evaluated by visual observation. Segregation patterns were evaluated for the “goodness of fit” to the expected ratios using the chi-squared test.

### 4.2. Sampling for Measurement of Leaf Senescence Related Physiological Parameters and RNA Extraction

Plants of HD and *yl1* were grown in the field in Zhucheng City, Shandong Province, during the 2018 growing season. Middle position leaves of each plant were collected at different developmental stages. The collection of the leaf samples was done at four developmental stages during the growing season: 35, 50, 75, and 95 days after transplanting (DAT). For each sample, one leaf from the middle position was collected from three plants randomly. Each leaf was divided into two parts along the main vein. One half was wrapped in aluminum foil, immediately immersed into liquid nitrogen, and stored at -80 °C until it was used for extracting the RNA and soluble protein, while the other half was put in sample bag and stored in the ice box for chlorophyll content measurements.

### 4.3. Measurement of Leaf Senescence Related Physiological Parameters

Chlorophyll was extracted and quantified as described previously [58]. Briefly, 0.5 g of the leaf sample was immersed in 30 mL ethanol overnight in darkness until the leaves became completely pale. The absorbance at 649 nm and 665 nm of the extraction solution was quantified using a UV-Vis spectrophotometer (TECAN, Infinite M200, Salzburg, Austria).

Fluorescence of living leaves was measured using a portable modulated chlorophyll fluorometer according to the manufacturer’s instructions (Opti-Sciences, OS1p, Hudson, USA).

Extraction of the soluble protein in each sample was according to the manufacturer’s instruction (CWBIO, CW0885M, Beijing, China). Briefly, about 0.2 g of the frozen leaves material was homogenized in 1 mL extraction reagent containing a protease inhibitor cocktail. The amount of soluble proteins in each sample was measured using the BCA protein assay kit (CWBIO, CW0014S, Beijing, China) according to the manufacturer’s instruction.

### 4.4. RNA Extraction and qRT-PCR

The total RNA was extracted using the TRIzol reagent (Invitrogen, Carlsbad, USA). First-strand cDNA was synthesized from 1 μg of total RNA using the PrimeScript^TM^ RT reagent Kit with gDNA Eraser (TAKARA, RR047, Dalian, China) according to the manufacturer’s instruction. qRT-PCR was carried out on a Thermal Cycler Block 7500 (ABI, Waltham, USA) according to the manufacture of TB GreenTM Premix Ex TaqTM II (Tli RNaseH Plus) (TAKARA, RR820, Dalian, China). The means of three biological and technical replicates were analyzed. The *Actin* was used as a reference gene for internal control, while *SAG12* and *RBCS* were used as senescence markers. The primer sequences of the genes for qRT-PCR are shown in Table 2.

### 4.5. DNA Extraction and SSR Analysis

Genomic DNA was extracted using the Plant Genomic DNA Kit (TIANGEN, DP305, Beijing, China). The DNA concentration was measured using a NanoDrop 2000 spectrophotometer (Thermo Scientific, Waltham, USA) and adjusted to 50 ng/µL. Tobacco simple sequence repeat (SSR) markers were used to map *YL1* based on the high-density genetic map of tobacco [30]. PCR was performed in a Veriti 96-Well Thermal Cycler (ABI, Waltham, USA) following the previously described method [44]. Briefly, each reaction contained 7.5 μL of 2 x Taq PCR MasterMix (TIANGEN, KT211, Beijing, China), 50 ng template DNA, and 1 μL of 5 μM forward and reverse primers in a volume of 15 μL. The PCR program was as follows: pre-denaturation at 95 °C for 3 min, followed by 30 cycles of 15 s at 95 °C, 30 s at 50–60 °C (the annealing temperature of different primer pairs), and 30 s at 72 °C, and then a final extension at 72 °C for 5 min. After that, 6 μL PCR products were separated on 6% nondenaturing polyacrylamide gel and visualized with silver staining as previously described [60].

### 4.6. Linkage Map and Genetic Distance

The BC1F1 population ((G3 × *yl1*) × *yl1*) was used to obtain the markers linked to *YL1*. The polymorphic SSR markers between G3 and *yl1* were partially selected and used to screen a small number of recessive individuals of the BC1F1 population with the premature leaf senescence phenotype, as well as their parents. After confirming the linkage between one marker and *YL1*, all adjacent polymorphic SSR markers were run across the whole recessive individuals of the population. The linkage map based on the segregations between *YL1* and the markers was constructed by QTL Ici Mapping software V4.0 [61], and the threshold value of the logarithm of odd (LOD) was set at 3.0. Genetic distances were calculated and presented in Kosambi centiMorgans (cM).

### 4.7. Plant Hormone Treatments

Leaf discs from middle leaves of plants grown in greenhouse 50 DAT were used for plant hormone treatments. Detached leaves were cut into small discs (~1.5 cm in diameter) and floated on an incubation buffer (1/2 MS, 3 mM MES, and pH 5.8) with and without hormones in Petri dishes with filter paper. The samples were incubated at 28 °C under 24 h light in a growth chamber. The plant hormone was diluted to working concentrations using the incubation buffer. The working concentrations of the plant hormones are 100 µM for 1 - aminocyclopropane - 1 - carboxylic acid (ACC) and 50 µM for methyl jasmonate (MeJA) [62].

## Figures and Tables

**Figure 1 plants-08-00415-f001:**
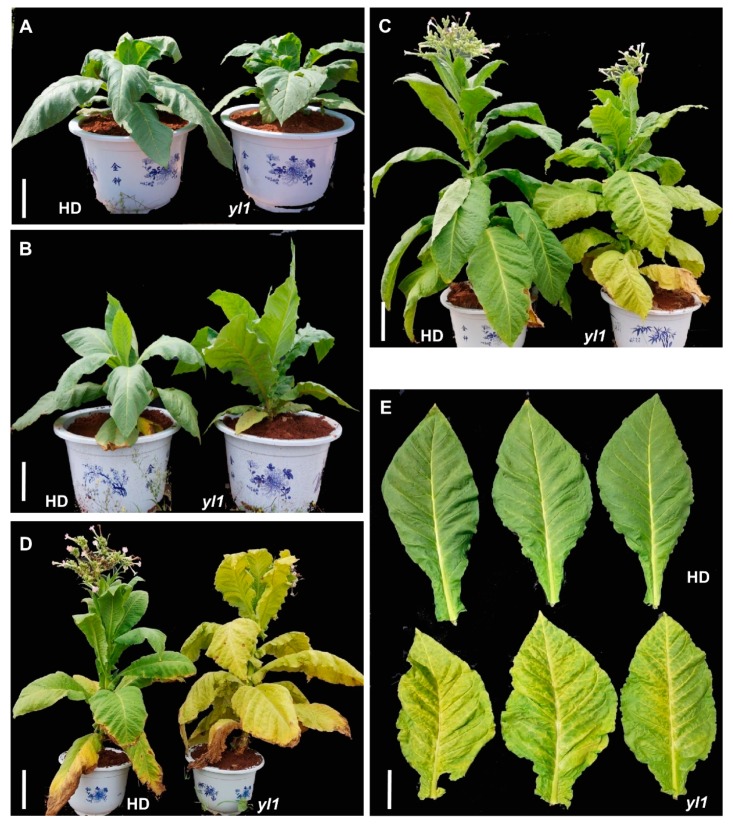
Phenotypic comparison between HonghuaDajinyuan (HD) and *yl1*. The phenotype of HD and *yl1* plants grown in the field. (**A**) 35 days after transplanting (DAT), bar 20 cm; (**B**) 50 DAT, bar 20 cm; (**C**) 75 DAT, bar 30 cm; and (**D**) 95 DAT, bar 35 cm. (**E**) The middle leaves of HD and *yl1* 75 DAT, bar 10 cm.

**Figure 2 plants-08-00415-f002:**
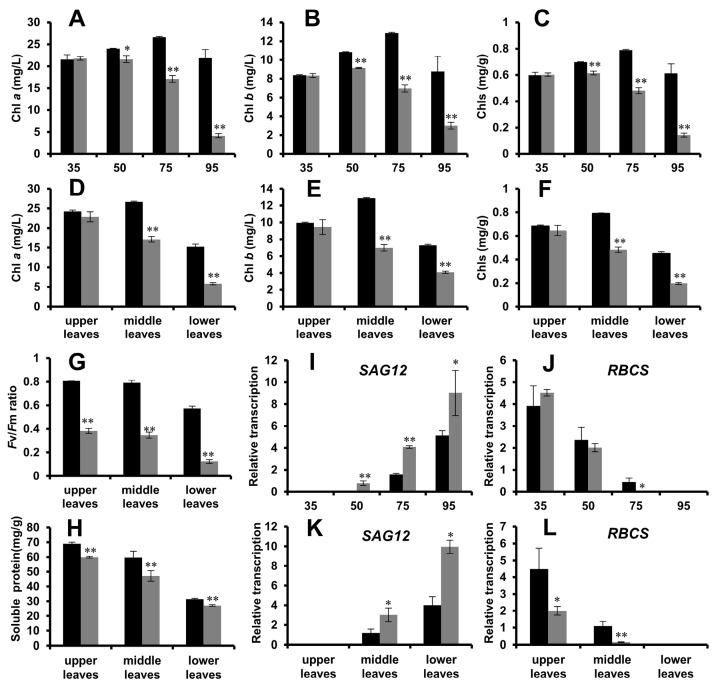
Leaf senescence related parameters of HD and *yl1*. Black and grey columns indicate HD and *yl1*, respectively. (**A**) Chlorophyll *a* (Chl *a*), (**B**) Chlorophyll *b* (Chl *b*), and (**C**) total Chlorophyll (Chls) in middle leaves of the same leaf position of HD and *yl1* at 35, 50, 75, and 95 DAT. (**D**) Chl *a*, (**E**) Chl *b*, (**F**) Chls, (**G**) *F*v/*F*m ratio, and (**H**) soluble protein in upper, middle, and lower leaves of the same leaf position of HD and *yl1* 75 DAT, respectively. Relative expression of (**I**) *SAG12* and (**J**) *RBCS* in middle leaves of the same leaf position of HD and *yl1* at 35, 50, 75, and 95 DAT. Relative expression of (**K**) *SAG12* and (**L**) *RBCS* in upper, middle, and lower leaves of the same leaf position of HD and *yl1* at 75 DAT. Values are mean ± SD of three individual replicates. * and ** represent significant difference determined by the Student’s t test *at p* ≤ 0.05 and *p* ≤ 0.01, respectively.

**Figure 3 plants-08-00415-f003:**
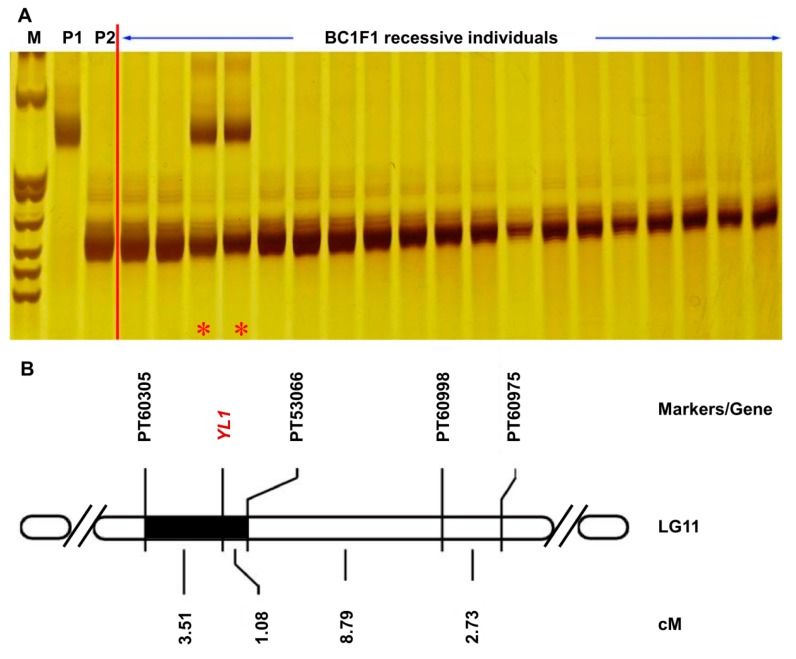
Preliminary mapping of *YL1*. (**A**) Linkage relationships between tobacco simple sequence repeat (SSR) markers PT53066 and *YL1* in 19 recessive individuals from BC1F1 (G3 × *yl1*) × *yl1* population. M indicates the DNA marker pBR322/Mspl; P1 indicates G3; P2 indicates *yl1*; and * indicates the recombinant individuals. (**B**) Genetic map of *YL1* and four SSR markers on tobacco linkage group 11 (LG11). The black bar indicates the smallest interval of *YL1* and markers; cM indicates centimorgan.

**Figure 4 plants-08-00415-f004:**
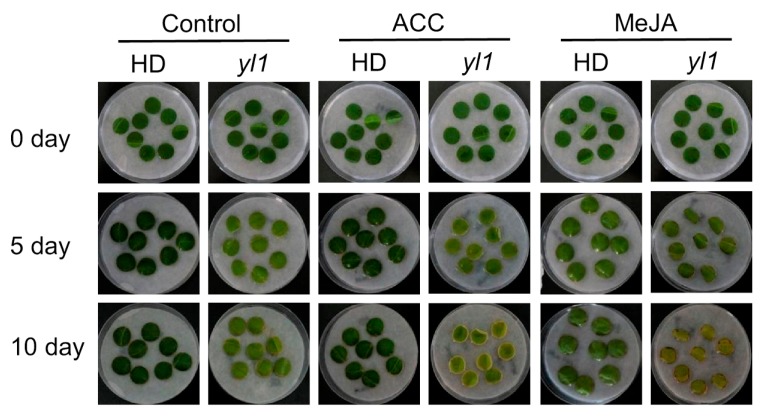
Plant hormone treatments of HD and *yl1* leaf discs. Controls were treated with the incubation buffer without hormone. ACC: 1 - aminocyclopropane - 1 - carboxylic acid; MeJA: methyl jasmonate.

**Table 1 plants-08-00415-t001:** Genetic analysis of *yl1*.

Population Type	Cross	Total	Wild Type	Mutant Type	Segregation Ration	χ^2 1^
F2	HD × *yl1*	154	111	43	2.581	0.701
F2	G3 × *yl1*	155	115	40	2.875	0.054
BC1F1	(HD × *yl1*) × *yl1*	155	83	72	1.153	0.781
BC1F1	(G3 × *yl1*) × *yl1*	163	77	86	0.895	0.497

HD: HonghuaDajinyuan, tobacco variety; G3: Gexin 3, tobacco variety; ^1^ Value for significant at 0.05 and *df* = 1 is 3.841.

**Table 2 plants-08-00415-t002:** The primer sequences of genes for qRT-PCR.

Gene Name	Accession Number ^1^	Forward Primer Sequence ^1^	Reverse Primer Sequence ^1^
*Actin*	Nitab4.5_0009320g0010.1	CAAGGAAATCACGGCTTTGG	AAGGGATGCGAGGATGGA
*SAG12*	Nitab4.5_0001608g0070.1	ATTTTCAGCGGTGGCAGCT	GTAAGAAGTCGTAGGCTCG
*RBCS*	Nitab4.5_0006249g0010.1	CCTGCTAAGGATACAATTAG	CTCAAATTTCTTGTTGTCA

^1^ The accession number and gene sequences were obtained from the *N. tabacum* CDS database (https://solgenomics.net) [59].

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
