# Peer review of "Characterization and Mapping of a Novel Premature Leaf Senescence Mutant in Common Tobacco (*Nicotiana tabacum* L.)"

_plants, 2019, doi:10.3390/plants8100415_

Round 1

Reviewer 1 Report

Dear Authors,

This manuscript investigated the function of YL1 in leaf senescence in tobacco. It demonstrates that yl1 upregulates senescence-associated genes and promotes age dependent leaf senescence by testing the expression of genes involved in senescence and chlorophyII contents. Furthermore, preliminary mapping of YL1 was performed. This study provides new information on the roles of  YL1 in leaf senescence in tobacco. Nonetheless, there are some issues needed to be further addressed and revised.

In the title - "Characterization" - However preliminary characterization has been performed in respect to the Tobacco yl1. I feel that data is not sufficient to characterize YL1 in tobacco properly.

It is possible that yl1 involved in age-dependent leaf senescence in tobacco. Your data in Figure1 supports it. However you have checked few parameters such as chlorophyll content and SAG12 senescence marker. Could you please add some data regarding photosynthesis parameters such as Photosynthetic  parameters at 75d and 95d?. In addition please improve the quality of figure 2.

Why lack of chlorophyll in yl1 mutant? due to defective chloroplast development? YL1 orthologs are known to be involved in chlloroplast development? Is it true with tobacco YL1? Please check some genes which are involved in chloroplast development.

Please check the subcellular localization of the YL1 protein. How much similarity is there when compared with Arabidopsis (At1g5200)?. How about the similarity compared to other orthologs such as Rice? Please discuss it.  

In figure 4, you have performed hormone treatments. Please add quantitative measurements such as chlorophyll content. Have you checked with ABA?

If you possible please do complimentation. Please describe the significance of yl1 in tobacco compared to other plants.

Thanks.

Reviewer 2 Report

The manuscript by Gao and colleagues presents a preliminary study of the yl1 mutant in tobacco. The research presented here was carried out correctly and the conclusions are supported by the experiments. The manuscript is mostly well written with a few edits required as listed below.

My question about this research is why did not the authors make use of the existing genome assemblies for Nicotinia such as assemblies for N. tabacum and N. benthamiana? While the current published assemblies for this genus may be fragmented and not organised in pseudo-chromosomes, it would be still worth it to determine where the SSR markers flanking the yl1 locus are located.  Furthermore, there is a high level of synteny in the Solanaceae: why not locating the flanking markers in the potato and tomato genomes? This could give some useful information about candidate genes, as well as testing the hypothesis of the role of genes involved in the ethylene and jasmonate pathways. This would be a simple experiment to do and would add a lot to the paper.

Minor edits:

Ln89: “marker” instead of “maker”.

Ln90: remove “preliminarily”

Ln90-93: this sentence does not belong to Introduction. Keep it for the Conclusion.

Ln160 and 161: I would use “linked markers” instead of “linkage”

Ln213: remove “the”. Michel et al located NtTPN1, …

Ln216: This sentence needs to be reworded. I think the authors mean that the number of SSR markers available means that it was not possible to further fine map the locus, but the current syntax is not correct.

Ln221: were 1.2 million SSRs really developed for Nicotinia, or were they detected in silico?

Round 2

Reviewer 1 Report

Dear Authors,

Figure 2 is ok.

Point 3, I have asked to add chlorophyll content. But you have not mentioned anything. Is it difficult?Y

you are now trying to clone the gene. It means that YL1 in tobacco is not fully characterized still. But preliminary characterization has been done.

Thanks.

Author Response

Point 1: I have asked to add chlorophyll content. But you have not mentioned anything. Is it difficult?Y

Response 1: We understand that the chlorophyll content may better show the result of hormone treatment. However, it is difficult to add this data in a short time for tobacco. The plants used in hormone treatment experiment are 50 days after transplanting. At present, we do not have plants in the developmental stage. We have just sowed seeds. It requires more time for repeating the hormone treatment and determining the chlorophyll content. The hormone treatment experiment was designed to speculate whether the causal gene is likely to be involved in the hormonal signalling pathway. In the present study, we mainly focused on the characterization of the premature leaf senescence mutant and mapping the causal gene of the mutant.

Round 3

Reviewer 1 Report

Dear Authors,

Your response is acceptable.

Thanks.